Soft anatomy and morphological variation in Daptomys peruviensis (Rodentia, Cricetidae), a rare ichthyomyine from the northwestern Amazonian forests

Brito Jorge jorgeyakuma@yahoo.es 1
Vega-Yánez Mateo A. 1 2 3
Guaya-Ramos Jhandry P. 1
Polo Melanie 1
Koch Claudia 4
Tinoco Nicolás 1 5
Pardiñas Ulyses F.J. 1 6
1 Instituto Nacional de Biodiversidad , Quito , Ecuador
2 Colegio de Ciencias Biológicas y Ambientales, Laboratorio de Biología Evolutiva, Campus Cumbayá, Universidad San Francisco de Quito , Quito , Ecuador
3 Colegio de Ciencias Biológicas y Ambientales, Maestría en Ecología Tropical y Conservación, Universidad San Francisco de Quito , Quito , Ecuador
4 Leibniz Institute for the Analysis of Biodiversity Change, Museum Koenig , Bonn , Germany
5 Sección de Mastozoología, Museo de Zoología, Facultad de Ciencias Exactas y Naturales, Pontificia Universidad Católica del Ecuador , Quito , Ecuador
6 Instituto de Diversidad y Evolución Austral , Puerto Madryn , Argentina
Brygadyrenko Viktor
Electronic publication date: 2025 Mar 5
Publication date: 2025
Volume: 13
Electronic Location ID: e18997
Received 2024 Nov 14; Accepted 2025 Jan 24
Copyright: ©2025 Brito et al.
Copyright year: 2025
Copyright holder: Brito et al.
License: This is an open access article distributed under the terms of the Creative Commons Attribution License, which permits unrestricted use, distribution, reproduction and adaptation in any medium and for any purpose provided that it is properly attributed. For attribution, the original author(s), title, publication source (PeerJ) and either DOI or URL of the article must be cited.
License URL: https://creativecommons.org/licenses/by/4.0/

Keywords: Ecuador, Ichthyomyini, Neusticomys, Sigmodontinae, Tropical Andes

Funding: The EcoMinga Foundation The Research Grant 2024 of the Master in Tropical Ecology and Conservation of the Universidad San Francisco de Quito Agencia Grant 2020-2068 This work was funded by the EcoMinga Foundation (Jorge Brito), the Research Grant 2024 of the Master in Tropical Ecology and Conservation of the Universidad San Francisco de Quito (Mateo A. Vega-Yánez), and Agencia Grant 2020-2068 (Ulyses F.J. Pardiñas). The funders had no role in study design, data collection and analysis, decision to publish, or preparation of the manuscript.

==============================
The recently resurrected genus Daptomys Anthony, 1929 includes poorly known small cricetid rodents that are widely distributed in tropical South America. Along with Neusticomys Anthony, 1921, these species are the most terrestrial members of the tribe, which is otherwise distinguished by adaptations that allow species to live in both aquatic and terrestrial environments. Newly collected Ecuadorean specimens provide complementary information of the craniodental and soft anatomy of Daptomys, focusing on rhinarium morphology, soft palate, stomach, caecum configuration, and other features. In addition, the phylogeny presented here, combined with species distribution models, suggests a simplified taxonomy indicating that Daptomys peruviensis (Musser & Gardner, 1974) has a wide distribution extending from Venezuela to Peru. In this novel scenario, Daptomys mussoi (Ochoa & Soriano, 1991) would be a junior synonym of D. peruviensis, and the application of a trinominal taxonomy appears premature.

Introduction

Among the lesser-known members of the specialized tribe Ichthyomyini, which includes living sigmodontines with unusual adaptations for preying on small animals by diving in rapid freshwater environments (e.g., Voss, 1988; Voss, 2015; Salazar-Bravo et al., 2023), are those classified in Daptomys Anthony, 1929. This genus, recently resurrected from the synonymy of Neusticomys Anthony, 1921, supposedly encompasses six species (Salazar-Bravo et al., 2023). These small sigmodontines, weighing less than 50 grams as adults, are widely distributed in eastern Andean forests and the Amazon Basin. Their more generalized body morphology, which reflects fewer adaptations to an aquatic lifestyle than seen in other tribal members, may explain this distribution. Additionally, ichthyomyines apparently have been restricted to the region north of the northern Andes for about half of their evolutionary history, and their wide geographic and environmental range suggests a relatively recent diversification event (Salazar-Bravo et al., 2023).

Of the various named and unnamed forms attributed to Daptomys, D. peruviensis (Musser & Gardner, 1974) is one of the least known. It is restricted to fewer than 10 recorded localities, mostly in Peru (Musser & Gardner, 1974; Solari et al., 2006; Medina et al., 2015; Gonzales, Arce-Merma & Zeballos, 2017; Pacheco et al., 2020). Recently, its taxonomy was revised to include the trinominal designation D. peruviensis musseri Pacheco et al., 2020, distinguishing a sample from northeastern Amazonian Peru (Pacheco et al., 2020). Despite this systematic addition and the description of several other congeneric, almost nothing beyond classical external and craniodental morphology is known about these mice.

Here, we report new specimens from Ecuador, confirming and extending the known geographical range of D. peruviensis within the country (Tirira, Reid & Engstrom, 2018; Salazar-Bravo et al., 2023). The study of these materials has allowed for the first description of several traits related to soft anatomy, as well as novel details of craniodental morphology obtained through non-invasive techniques. Furthermore, the resolved phylogenetic position of these samples, along with potential distribution models, highlights the necessity to adjust the current classificatory scheme for this and other entities within the genus.

Materials & Methods

Studied specimens

The new Ecuadorean specimens studied here were collected during field trapping expeditions conducted in 2022 and 2023 by the senior author and collaborators at the localities of Piatua (1°12′18.32″S, 77°57′10.22″W, 813 m, Provincia de Pastaza; geographical coordinates with WGS84 datum recorded by GPS at the trap line) and Chawalyaku (1°30′37.87″S, 78°7′21.00″W, 1,175 m, Provincia de Morona Santiago). The animals were obtained from pitfall traps set in a dense premontane forest. At each locality, two lines of 10 buckets (20 liters each) were established. These lines remained active for 10 consecutive days, totaling 400 pitfall trap nights. Research permits were obtained from the Ministerio del Ambiente, Agua y Transición Ecológica de Ecuador (scientific research authorization No MAATE-ARSFC-2022-2583, and MAATE-DBI-CM-2023-0334). The collected animals were euthanized following standard procedures (Sikes, 2016), measured fresh, and preserved as carcasses with skins and skeletons cleaned and housed at the mammal collection of the Instituto Nacional de Biodiversidad (INABIO, Quito, Ecuador; acronym MECN). In addition, the specimen MECN 6629 from the Cordillera del Cóndor (3°45′25.78″S, 78°30′3.80″W, 1,406 m) reported as Daptomys sp. in Salazar-Bravo et al. (2023) and Pardiñas et al. (2024), was included in the analyzed samples for this contribution.

The novel Ecuadorean material was identified as Daptomys peruviensis after genetic confirmation (see below). The morphological analysis of these individuals followed the concepts described by Voss (1988), Carleton & Musser (1989), Pacheco et al. (2020), and Salazar-Bravo et al. (2023) for general external and craniodental anatomy; Pardiñas et al. (2024) for rhinarium; Quay (1954) and Carleton (1980) for soft palate; Carleton (1973) for stomach; and Vorontsov (1967) for caecum. External and craniodental measurements of the new specimens were taken according to the dimensions explained in Pacheco et al. (2020) to ensure the comparability of the results.

X-ray micro CT

Detailed information on the cranial and mandibular morphology was obtained through a high-resolution X-ray micro-computed tomography scan (micro-CT) of the skull of MECN 8072 with a Bruker SkyScan 1173 (Bruker MicroCT, Kontich, Belgium) at the Leibniz Institute for the Analysis of Biodiversity Change, Museum Koenig (LIB, Bonn, Germany). To avoid movements during scanning, the skull was placed in a small plastic container and embedded in cotton wool. Acquisition parameters comprised: An X-ray beam with a source voltage of 53 kV and a current of 124 µA, without the use of a filter; 960 projections of 600 ms exposure time each with a frame averaging of 4; rotation steps of 0.25° recorded over a 180° continuous rotation, resulting in a scan duration of 58 min; and a magnification setup generating data with an isotropic voxel size of 15.97 µm. The CT-dataset was reconstructed with N-Recon software version 1.7.1.6 (Bruker MicroCT, Kontich, Belgium) and rendered in three dimensions using CTVox for Windows 64 bits version 3.0.0 r1114 (Bruker MicroCT, Kontich, Belgium).

Molecular data

DNA extraction, PCR amplification, and sequencing with Oxford Nanopore Technologies were performed at the Nucleic Acid Sequencing Laboratory belonging to the INABIO. DNA was extracted from liver samples using the GeneJET Genomic DNA Purification Kit (K0722). Amplification was achieved through Polymerase Chain Reaction (PCR) using the primers MVZ05 and MVZ16 (Smith & Patton, 1993), and the GoTaq® Green Master Mix 2X kit to amplify the cytochrome b (Cyt b) sequence. PCR conditions included initial denaturation at 95 °C for 2 min, followed by 35 cycles of 95 °C for 30 s, 45 °C for 30 s, and 72 °C for 80 s, with a final extension at 72 °C for 5 min. The expected amplicon length was approximately 1,200 bp. The mitochondrial Cyt b marker was sequenced using a MinION Mk1C with Flongle Flow Cells R10.4.1 and the Rapid Barcoding Kit 96 (SQK-RBK114.96), following standard protocols. Data was high-accuracy (HAC) basecalled. The resulting FASTQ files were filtered at a Q score of nine, and consensus sequences were produced with NGSpeciesID (Sahlin, Lim & Prost, 2021). A total of 36 sequences were used for phylogenetic analysis. Available Cyt b sequences of closely related individuals and additional outgroup sequences were obtained from GenBank (http://www.ncbi.nlm.nih.gov/genbank/). The alignment was performed using the MAFFT algorithm in Mesquite version 3.81 (Maddison & Maddison, 2023) to edit and concatenate alignments. A maximum-likelihood tree was generated in IQ-TREE (Trifinopoulos et al., 2016) under default settings. Genetic distances for Daptomys were calculated in MEGA 11 (Kimura, 1980) based on the Kimura 2-parameter model.

Species distribution model

We retrieved 11 recording localities of D. peruviensis from literature (Musser & Gardner, 1974; Pacheco & Vivar, 1996; Gonzales, Arce-Merma & Zeballos, 2017; Tirira, Reid & Engstrom, 2018; Salazar-Bravo et al., 2023), which were geographically validated, in addition to the three new records from Ecuador (i.e., Chawalyaku, Cordillera del Cóndor, and Piatua) that were included in the species distribution modeling (Supplemental S1). The bioclimatic variables were downloaded from WorldClim v2.1 at a spatial resolution of approximately 2.5 min (Fick & Hijmans, 2017). The variables used in the models were selected with the R package usdm (Naimi et al., 2014) based on the variance inflation factor (VIF), using a threshold of 0.7 to reduce collinearity between variables (Marquardt, 1970; De Marco & Nóbrega, 2018). According to that analysis, the following variables were selected: bio2 (mean diurnal range (mean of monthly (max temp–min temp))), bio3 (Isothermality (bio2/bio7) (× 100)), bio4 (Temperature seasonality (standard deviation × 100)), bio8 (Mean temperature of wettest quarter), bio13 (Precipitation of wettest month), bio14 (Precipitation of driest month), bio18 (Precipitation of warmest quarter), and bio19 (Precipitation of coldest quarter). Moreover, the occurrence data were randomly divided into two sets: 70% for calibration and 30% for validation. Additionally, using the ecospat package (Broennimann, Di Cola & Guisan, 2023), we generated pseudo-absences for training and test occurrences following Castelblanco-Martínez et al. (2021). These pseudo-absences were created within the area M (Soberon & Peterson, 2005), which in our case corresponds to the Amazon River basin. With the R package sdm (Naimi & Araujo, 2016) we used six modeling algorithms: Classification and Regression Trees (CART), generalized additive model (GAM), generalized linear model (GLM), maximum entropy (Maxent), Random Forest (RF), and support vector machine (SVM). For the models, five-folds of cross-validation, 10 bootstrapping and 10 replicates were generated for each algorithm. Finally, each model was evaluated based on true skill statistic (TSS) and omission rate (Allouche, Tsoar & Kadmon, 2006; Li & Guo, 2013) to build a consensus ensemble model of D. peruviensis.

Results and Discussion

Genetic (cytb) structure and distances

The three new Ecuadorean specimens were closely grouped with two sequences of D. peruviensis musseri from Loreto (Peru). This clade was sister to a sequence of Daptomys mussoi (Ochoa & Soriano, 1991) from Colombia, and both taxa were sister to D. peruviensis peruviensis (animals from Cusco and Ucayali). This composite clade (i.e., D. peruviensis plus D. mussoi) was resolved as sister to the Amazon species D. ferreirai (Percequillo, Carmignotto & Silva, 2005), and this clade was sister to the Guianan D. oyapocki Dubost & Petter, 1979. Finally, all these species were resolved as sister to a Bolivian sequence belonging to an unnamed taxon (Salazar-Bravo et al., 2023). The monophyly of the genus was highly supported (Fig. 1A). Genetic distances within D. peruviensis ranged from 0.09% to 2.62%; the lowest was between the specimens MUSA 19658 (Cusco, Peru) and MUSA 12657 (Ucayali, Peru); and the highest was between MUSA 19658 (Cusco, Peru) and MUSM 45735 (Loreto, Peru). The distance between D. peruviensis and D. mussoi varied from 1.23% to 1.92% (Fig. 1B).

Figure 1 Phylogeny of Daptomys.

Phylogenetic relationships of Daptomys within the Ichthyomyini (A), and genetic distances for the Daptomys of Colombia, Ecuador, and Peru (B). Abbreviations employed in terminal labels: BO, Bolivia; BR, Brazil; CO, Colombia; EC, Ecuador; GU,Guyana; PE, Peru.

Geographic distribution

D. peruviensis occurs from the eastern foothills of the Andes in south-central Peru, including the Amazon lowlands in Loreto (Musser & Gardner, 1974; Solari et al., 2006; Medina et al., 2015; Gonzales, Arce-Merma & Zeballos, 2017; Pacheco et al., 2020), through the Amazon and eastern foothills of Ecuador (Tirira, Reid & Engstrom, 2018; Salazar-Bravo et al., 2023), to the border between Colombia and Venezuela (Ochoa & Soriano, 1991; Salazar-Bravo et al., 2023; Fig. 2). Its altitudinal occurrence ranges between 110 and 1,400 m.

Figure 2 Species distribution model.

Consensus ensemble model for habitat suitability of Daptomys peruviensis in the Amazon River basin (red dots are new Ecuadorean occurrences). Localities: 1 = Villa Carmen, 2 = Aguas Calientes, 3 = Pakitza, 4 = Río Curanja, 5 = Río Shesha, 6 = Loreto, 7 = Cordillera del Cóndor, 8 = Chalwayaku, 9 = Piatúa, 10 = Villano, 11 = Yasuní, 12 = Vereda San Carlos, 13 = Vereda Rosa Blanca, 14 = Río Potosí. Model developed by Mateo A. Vega-Yánez.

Morphology

According to Pacheco et al. (2020), several traits distinguish Daptomys peruviensis from other congeneric species, including pelage color, body size, morphology of the zygomatic plate and incisive foramina (Fig. 3), development of the Eustachian tube, ventral emargination of the foramen magnum, and cranial profile (see also Musser & Gardner, 1974; Voss, 1988). The new material from Ecuador conforms to the morphological diagnoses characters of Ichthyomyini and Daptomys in Salazar-Bravo et al. (2023), and largely conforms to these diagnostic characters of D. p. musseri. However, the Ecuadorian specimens also exhibit characteristics previously considered exclusive to D. p. peruviensis (see Pacheco et al., 2020), such as a relatively long and broad Eustachian tube (Figs. 3 and 4). Additionally, craniodental measurements show partial overlap between these forms and D. mussoi (Table 1).

Figure 3 Skull and molar series.

Reconstructed CT scan images of the cranium in dorsal (A), ventral (B), and lateral (C) views, and mandible in labial view (D) and photographs of the right upper (E) and left lower molar series (F) of Daptomys peruviensis (MECN 8072; Piatúa, Pastaza, Ecuador (A–D); MECN 7175, Chawalyaku, Morona Santiago, Ecuador (E–F)). Scale bar: short five mm, long one mm. Three-dimensional reconstruction by C. Koch and J. Brito.

Figure 4 Views of the auditory bulla and foramen magnum region.

Reconstructed CT scan images (ventral views) of the right auditory (ab) with a large Eustachian tube (et; A), and the triangular ventral margin of the foramen magnum with small lateral processes (lt; B) of Daptomys peruviensis (MECN 8072; Piatúa, Pastaza, Ecuador). Three-dimensional reconstruction by J. Brito.

Table 1 External and craniodental measurements of adult specimens of Daptomys peruviensis including two specimens from Ecuador.

All measurements are in millimeters, except for the mass body, which is expressed in grams. Ranges are followed by sample size (in parentheses). Abbreviations are: m, male; f, female.

	D. mussoi	D. p. peruviensis	D. p. musseri	MECN 6629	MECN 8072	
	1 m, 1 f	2 m	2 m	m	m	
Head-body length	94–118 (2)	128–133 (2)	117–126 (2)	96	123	
Tail length	81–82(2)	108–110 (2)	82–85 (2)	83	97	
Hindfoot length	21(2)	27.4–30.0 (2)	24.0–27.0 (2)	20	28	
Ear length	10(2)	11.4–12.0 (2)	9.0–11.5 (2)	9	12	
Body mass			40–49 (2)	26	34	
Condyloincisive length	24.1–24.7 (2)	28.2–29.5 (2)	26.5–27.2 (2)	24.3	27.8	
Length of nasals	9.5 (1)	10.7–11.1 (2)	10.3 (2)	9.1	11.3	
Length of diastema	6.2(2)	7.1–7.7 (2)	7.5–7.9 (2)	6.9	7.3	
Length of the incisive foramina	4.6–4.8 (2)	5.7–5.8 (2)	5.3–5.1 (2)	4.5	5.1	
Length of the maxillary molars	3.3–3.4 (2)	3.9–4.2 (2)	3.6 (2)	3.6	3.5	
Breadth of nasals	3.3 (2)	3.7–3.8 (2)	3.3–3.5 (2)	3.1	3.3	
breadth of the incisive foramina	1.8 (2)	2.0 (2)	2.2–2.4 (2)	1.7	1.7	
Breadth of the palatal bridge	2.5–2.6 (2)	2.7 (2)	3.0–3.1 (2)	2.1	2.6	
Breadth of M1	1.2 (2)	1.3 (2)	1.2–1.3 (2)	1.3	1.3	
Least interorbital breadth	4.5–4.6 (2)	5.1 (2)	5.2–5.3 (2)	4.8	5.1	
Zygomatic breadth	12.2–12.3 (2)	14.6–15.4 (2)	14.0–15.1 (2)	12.4	14.2	
Breadth of braincase	11.0 (2)	12.8 (2)	11.9 (2)	11.6	12.6	
Breadth of the zygomatic plate	1.1 (2)	1.3 (2)	1.0–1.1 (2)	1.2	1.3	
Breadth of the incisor tips	1.6–1.9 (2)	2.2–2.3 (2)	1.9–2.1 (2)	1.6	1.9	
Height of incisors	4.1–4.3 (2)	4.9–5.1 (2)	5.3–5.7 (2)	4.3	4.5	
Depth of incisors	1.5 (2)	1.8–1.9 (2)	1.7–1.9 (2)	1.4	1.6	
Breadth of the occipital condyles	6.2–6.5 (2)	7.5–7.6 (2)	6.8–7.1 (2)	7.3	7.4	
Source	Ochoa & Soriano (1991)	Musser & Gardner (1974) and Medina et al. (2015)	Pacheco et al. (2020)	This study	This study	

The Ecuadorian specimens are robust, stocky animals with dense fur and prominent mystacial vibrissae, and they exhibit an overall brownish coloration (Fig. 5), consistent with previous descriptions of D. peruviensis. The manus and pes are sparsely haired, and there are small webs among digits II, III, and IV on the pes. The metacarpal configuration is: III > IV > II >> V >> I; and the metatarsal configuration is: III > IV > II >> V > I. Hind feet show plantar squamation; dark brown plantar surface; a rudimentary tubercle resembling the hypothenar is present in the young specimen, while in the adult it is absent (Fig. 6A). Fringing hairs on pes are weakly developed. The ungual tufts are short, barely reaching the middle of the claws (Figs. 6A–6B).

Figure 5 External appearances of Daptomys peruviensis in its natural habitat in Piatúa, Pastaza, Ecuador (MECN 8072).

Photographs by J. Guaya.

Figure 6 External and soft morphology of Daptomys peruviensis: ventral (A, C) and dorsal (B, D) aspect of the right pes (A, B) and manus (C, D); upper lips in frontal view (E), rhinarium (F) and soft palate (G). The specimens figured are MECN 8072: Piatúa, Pastaza, Ecuador (A, B); MECN 6629: Cordillera del Cóndor, Zamora Chinchipe, Ecuador (C–G).

Abbreviations: I–V, digits; d1–d3, diastemal rugae; i1–13, interdental rugae; p, philtrum; dn, dorsum nasi; H, tubercle of Hill; sm, sulcus medianus; t, tubercle hypothenar. Photographs by J. Brito.

In these Ecuadorean specimens, the upper lips are densely furred, showing no distinction from the rest of the muzzle. The mystacial vibrissae are dense, rigid and long, extending beyond pinnae when laid backward. Genal and superciliary vibrissae are absent. The philtrum is a bare, well-defined groove, featuring a prominent upper cleft (Fig. 6E). The rhinarium exhibits the “cherry” pattern, a configuration observed in other members of the tribe (Pardiñas et al., 2024); it consists of an enlarged, bare dorsum nasi, continuous with the tubercle of Hill; the areola circularis is faintly distinguishable, though present and large, covering most of the exposed surface; the epidermal ridges (or rhinoglyphics) are nearly invisible to the naked eye, but appear as narrow, transverse lines aligned with the main axis of the areola; and a sulcus medianus divides the rhinarium, extending across the dorsum nasi and bisecting the two halves of the tubercle of Hill (Fig. 6F).

The soft palate consists of three diastemal and three interdental rugae. The anteriormost ridge is fused to the posterior portion of the incisive papilla, forming a broad epithelial mound that extends posteriorly as a lower medial ridge, connecting to the next ruga. The second diastemal ruga is straight and exhibits a medial projection, forming a low relief that contacts the third diastemal ruga, which arches forward at this point. Additionally, between these two diastemal rugae, clearly defined labial paired mounds resemble incomplete ridges. This arrangement of diastemal rugae gives this part of the soft palate the appearance of two contiguous chambers externally closed by fleshy cingula. In contrast, the interdental rugae are notably flat, short, and broad, with a wide medial space between them (Fig. 6G). Voss (1988) described the presence of three diastemal rugae as the generalized condition in ichthyomyines, but noted four interdental rugae in the single Daptomys he examined, the type species of the genus, D. venezuelae Anthony, 1929.

The stomach (30 mm width, 25 mm height) conforms to the unilocular-hemiglandular condition, which is common in most described sigmodontines (e.g., Carleton, 1973; Pardiñas et al., 2020). It appears well-muscled, with the corpus developed into an expanded fornix ventricularis and a comparatively small antrum (Figs. 7A–7C). In the antrum, the glandular epithelium covers most of the region, with a broad bordering fold arched to the left, extending beyond the esophageal opening. The cornified epithelium has a surface lacking an enterolitic appearance, and the pyloric pars is indistinct, while both the incisura angularis and cardialis are very shallow. The poorly preserved stomach of specimen MECN 6629, illustrated in Salazar-Bravo et al. (2023: fig. 39A) as Daptomys sp., displays the same basic features described above. The stomach morphology of D. peruviensis matches that of the only previously studied species in the genus, D. venezuelae, first analyzed by Carleton (1973) and redrawn by Voss (1988). If the antrum and corpus are recognized as distinct regions separated by an imaginary line running from the esophageal opening to the fundus in a perpendicular orientation, in D. peruviensis the glandular epithelium is mainly restricted to the antrum, closely resembling Carleton’s (1973: fig. 6A) drawing of D. venezuelae.

Figure 7 Digestive system of Daptomys peruviensis.

Gross morphology of partial digestive system of Daptomys peruviensis (MECN 8072; Piatúa, Pastaza, Ecuador): from stomach to rectum (A); stomach in dorsal (B) and internal (C) views; and caecum (D). Abbreviations: b, bordering fold; co, cornified epithelium; d, duodenum; ge, glandular epithelium; i, incisura angularis. Scale = 10 mm. Photographs by J. Brito.

The small intestine is long (approximately 350 mm) and almost continuous with the short large intestine (around 45 mm). The caecum appears very simple, lacking an appendix and chambers, with the proximal colon being parallel or U-shaped (sensu Behmann, 1973). There are mostly no previous descriptions of the caecum in ichthyomyines, even in the two major works on the tribe (i.e., Voss, 1988; Salazar-Bravo et al., 2023). Paradoxically, the only specific description (and illustration) was provided much earlier by Thomas (1893: 337) when he established Ichthyomys, noting that the “Caecum…much reduced in volume, very short, and only of the same diameter as the rectum”. The general features of the caecum in Daptomys and Ichthyomys are consistent with those expected in muroids adapted to highly proteinaceous diets (e.g., Vorontsov, 1982; Langer, 2017).

In the Ecuadorian specimens of D. peruviensis, the tuberculum of the first rib articulates with the transverse processes of both the seventh cervical and first thoracic vertebrae. The second through fourth thoracic vertebrae have small, similar-sized neural spines. Regarding the latter, Voss (1988) observed that all ichthyomyines differ from other sigmodontines in the site of attachment for the nuchal ligament and its corresponding osteological marker, an enlarged neural spine. While most sigmodontines show an enlarged neural spine on the second thoracic vertebra (cf. Carleton, 1980), ichthyomyines have this attachment on the third thoracic vertebra. In D. peruviensis, the second thoracic vertebra has a neural spine that is not enlarged, and the third thoracic vertebra has a neural spine that is not differentiated in size from the others, although its apical portion is slightly rounded (Fig. 8).

Figure 8 View of the axial skeleton.

Lateral view of the axial skeleton including from axis to eighth thoracic vertebra and associated ribs and sternebrae of Daptomys peruviensis (MECN 8072; Piatúa, Pastaza, Ecuador). Abbreviations: C3–C7, third to seventh cervical vertebrae; np, neural process; T1, T2, T3 and T4, first to fourth thoracic vertebrae. Scale = 10 mm. Photographs by J. Brito.

The vertebral column of D. peruviensis (based on the MECN 8072) consists of 19 thoracicolumbar vertebrae, four sacral vertebrae (the first three fused), and 26 caudal vertebrae, with a total of 13 ribs. These values differ from those previously reported for D. venezuelae, which has 30 to 33 caudal vertebrae and 14 ribs (Voss, 1988).

Species distribution models

The species distribution model (SDM) shows highly suitable areas for D. peruviensis habitat within the Amazon River basin (Fig. 2). These areas include the Yungas in Bolivia, Madre de Dios in Peru, the western region of the Brazilian Amazon bordering Peru, Northern Amazon of Ecuador, Parque Nacional Natural Yaigojé Apaporis, Cordillera Central, Sierra Nevada de Santa Marta, and Puerto Santander in Colombia, and the Mérida Mountain range in Venezuela. The SDM also identified areas within the Amazon River basin with a probability of occurrence of less than 0.2 that agrees with previously constructed models (Pacheco et al., 2020).

Final considerations

Daptomys peruviensis was described 50 years ago (Musser & Gardner, 1974). To date, only 14 individuals have been recorded in Peru, Ecuador, Colombia, and Venezuela, including the three new Ecuadorian specimens presented here (e.g., Pacheco et al., 2020). Several factors may explain the limited number of recorded individuals, including the species’ strong preference for primary evergreen lowland forests, which it avoids in areas impacted by human activities (Voss, 2015). This specialization severely restricts the locations where researchers can find it. Additionally, the difficulty of capturing of D. peruviensis may be a significant factor, leading to the scarcity of museum specimens (Pacheco et al., 2020; this study).

The SDM estimates that the species occurs in a unique pattern with population patches scattered throughout the western Amazon. If confirmed, this pattern could reflect a historically widespread range fragmented by changing Quaternary climatic conditions (e.g., Haberle & Maslin, 1999; Thom et al., 2020). These findings underscore the importance of SDM as an essential tool in biogeography, allowing the study of species distribution patterns across different spatial and temporal scales (Araújo & Peterson, 2012; Trappes, 2021). Moreover, the case of D. peruviensis highlights the need for increased trapping efforts, not only to refine these models but also to provide direct materials for various studies.

The recently proposed trinominal classification of D. peruviensis (Pacheco et al., 2020) may be premature. Given the limited number of specimens available in collections, the degree of morphological variation observed, and the challenges in testing population variability, it might be more appropriate to revert to a monotypic treatment of D. peruviensis. Voss (2015) discusses the close relationship between D. peruviensis and D. mussoi, suggesting they may be conspecific. The findings presented here support the hypothesis that D. mussoi is a junior synonym of D. peruviensis. A more comprehensive revision, including an examination of the entire genus, is necessary to address these issues.

If, in the future, as suggested by SDM, isolated populations are recognized as separate nuclei, and morphological traits demonstrate the existence of geographic races, then the names musseri and mussoi would be available to distinguish potential subspecies. With the evidence currently at hand, there is little to support anything more than geographic structure in the genetic data, which does not meet the criteria required for a trinominal proposal (cf. Patton & Conroy, 2017).

Supplemental Information

Supplemental Information 1 Genetic data matrix (Cytb) of the genus Daptomys and outgroups

Supplemental Information 2 Cytb sequence: MECN 6629, Cordillera del Cóndor, Zamora Chinchipe, Ecuador

Supplemental Information 3 Cytb sequence: MECN 8072, Piatua, Pastaza, Ecuador

Supplemental S1 Records SDM

Daptomys peruviensis record localities used for modeling

To Juan P. Reyes and Marco Monteros for their valuable help in the field work in Piatúa and Chawalyaku. Thanks to Benjamin & Yacine Ortiz from Chalwayaku Ecoaldea. To the members of the Colectivo Piatua Resiste: Jessica Grefa, Alexis Grefa, Julissa Alvarado, Octavio Grefa, Roger Grefa, Kayu Huatatoca, Karen Grefa, Elvia Grefa for all the support provided during the field phase. Thanks Daniela Franco-Mena (LBE-USFQ) for her valuable support in the phylogenetic component, to Julián A. Velasco for his important guidance in the SDMs, and to Rudolf Haslauer for his deep effort to obtain for us the dissertation of H. Behmann. Miguel Pinto helped with assistance with sequencing in GenBank. The critical comments provided by the reviewers Robert Voss and Aldo Caccavo helped to improve the correctness of the manuscript and are greatly appreciated. We used Deepl for grammar correction.

Additional Information and Declarations

Competing Interests

Author Contributions

Animal Ethics

Field Study Permissions

DNA Deposition

Data Availability

The authors declare there are no competing interests.

Jorge Brito conceived and designed the experiments, performed the experiments, analyzed the data, prepared figures and/or tables, authored or reviewed drafts of the article, and approved the final draft.

Mateo A. Vega-Yánez conceived and designed the experiments, performed the experiments, analyzed the data, prepared figures and/or tables, authored or reviewed drafts of the article, and approved the final draft.

Jhandry P. Guaya-Ramos analyzed the data, prepared figures and/or tables, authored or reviewed drafts of the article, and approved the final draft.

Melanie Polo conceived and designed the experiments, analyzed the data, prepared figures and/or tables, authored or reviewed drafts of the article, and approved the final draft.

Claudia Koch analyzed the data, prepared figures and/or tables, authored or reviewed drafts of the article, and approved the final draft.

Nicolás Tinoco analyzed the data, authored or reviewed drafts of the article, and approved the final draft.

Ulyses F.J. Pardiñas conceived and designed the experiments, performed the experiments, analyzed the data, prepared figures and/or tables, authored or reviewed drafts of the article, and approved the final draft.

The following information was supplied relating to ethical approvals (i.e., approving body and any reference numbers):

Handling and all activities regarding specimens followed care and use ethical procedures recommended by the American Society of Mammalogists (Sikes, 2016). For the use and care of animals, we follow the guidelines of the Ministerio del Ambiente, Agua y Transición Ecológica del Ecuador, through scientific research authorization No MAATE-ARSFC-2022-2583, and MAATE-DBI-CM-2023-0334.

The following information was supplied relating to field study approvals (i.e., approving body and any reference numbers):

Field experiments were approved by the Ministerio del Ambiente, Agua y Transición Ecológica del Ecuador (permissions number No MAATE-ARSFC-2022-2583).

The following information was supplied regarding the deposition of DNA sequences:

The sequences are available at GenBank: PQ587323 and PQ587324.

The following information was supplied regarding data availability:

The data are available in the Supplemental Files.

The media is available at MorphoBank: Project 5570, DOI: 10.7934/P5570.

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
