# Peer review of "Soft anatomy and morphological variation in Daptomys peruviensis (Rodentia, Cricetidae), a rare ichthyomyine from the northwestern Amazonian forests"

_PeerJ, doi:10.7717/peerj.18997_

## Round 0.1 · original submission · Minor Revisions

Dear authors, I ask you to correct the manuscript very carefully in accordance with all the recommendations of the reviewers. I hope that the new version of this article will be approved by the reviewers and will be published in our journal as soon as possible.

·

Basic reporting

.

Experimental design

.

Validity of the findings

.

Additional comments

This is a useful summary of new information about the widespread but seldom collected ichthyomyine species Daptomys peruviensis. The anatomical character data are potentially useful for any future analysis of phenotypic evolution in this interesting adaptive radiation of semiaquatic carnivorous mice, but I made numerous suggestions for minor improvements in the text using TrackChanges. A statement to the effect that, except as noted, the examined specimens conform to the morphological diagnoses of Ichthyomyini and Daptomys in Salazar-Bravo et al. (2023) would be useful.

·

Basic reporting

The present manuscript increases the current understanding about morphological and molecular variation within Daptomys peruviensis based on recently collected specimens from Ecuador, which expanded the known distribution of this species. It is clear and well written and both introduction and background sufficiently contextualize the relevance of the findings reported here. The text is mostly well structured; however, the authors combined the results and discussion sections, which does not conform with PeerJ standards. Figures are relevant and with adequate quality. Highlighting, in Fig 6 and Fig 7, the morphological features used in the text description of rhinarium, palate and intestines would surely improve comprehension of these section. Raw data is adequately supplied as supplemental Information.

Experimental design

The objectives of the present manuscript are well defined and sufficiently covered within the text. Methods are explicit and well described. Most of the results presented in this manuscript are based on descriptions accompanied by well-elaborated figures, which allow the evaluation and replication of the data discussed in the text. Molecular alignment and specimen sequences, that support the analysis used in the present manuscript are also supplied. Concerning the Species Distribution Models, although the localities are numbered and briefly described in the Results, it would be advisable to organize and include the localities, geographic coordinates and the sources in a table.

Validity of the findings

In general, results reported within the manuscript are clear, well supported and improve the current knowledge on Daptomys peruviensis, specially regarding Peruvian and Ecuadoran specimens. However, through the text, it seems that the authors assumed that D. mussoi is not a valid species, mainly based on the results of the molecular phylogeny, which seemed to be supported by the Species Distribution Models. Although the placement of the Colombian individual associated with the name D. mussoi falls within the range of variation of D. peruviensis, a taxonomic proposal of synonymy might be precipitated because: 1) the analysis reported here does not include the type or specimens from the type locality, which are directly associated with the name D. mussoi, instead, the authors are generalizing that the genetic information of the Colombian specimen represents the Venezuelan specimens, rather than adopting a more conservative approach that considers the possibility of the Colombian specimen (or population) being misidentified; 2) The molecular pattern recovered is based only on one mitochondrial marker, and the analysis of other genetic markers could exhibit a different result. Furthermore, morphological and molecular information regarding D. mussoi is disconnected, as the morphological information of the sequenced specimen is not explored in the present manuscript. I recommend a more conservative approach of this issue, allocating the Colombian populations to D. peruviensis but exercising caution about the Venezuelan populations, until a proper review of these samples is accomplished (based on morphological data) or until the holotype or at least another specimen of the type locality (or a near locality) is included in the molecular analysis.

Additional comments

This manuscript makes significant contributions oh the knowledge available for Daptomys peruviensis, detailing several morphological structures, including soft tissues, and offering both a hypothesis of phylogenetic relationship among different populations of D. peruviensis and a model of potential distribution. This model could help increase the availability of specimens of D. peruviensis in collection through newly collected material. The manuscript is clear and well written but I really recommend that the authors consider the issues highlighted above in this revision before this study is published. For clarity, a list of the issues is given bellow:
1- Carefully consider any nomenclatural proposal within this work as the holotype is not included and the Venezuelan populations are not proper included in the available analysis;
2- Reorganize the text to include separated sections for Results and Discussion, following PeerJ standards.
3- Provide a table with details of the locations used to generate SDM, including coordinates and sources for each location;
4- Include symbols in Figures 6 and 7 to indicate morphological structures detailed in the text, to improve understanding of the description without requiring readers to be familiar with the specific nomenclatures.

---

## Round 0.2 · accepted · Accept

Dear authors, good afternoon. I am pleased to inform you that your article has been accepted for publication in our journal. I hope that you will continue your research on mammals in South America. Good luck to you!

·

Basic reporting

-

Experimental design

-

Validity of the findings

-

Additional comments

As I stated previously, the present manuscript represents an welcomed contribution to our knowledge about Daptomys, specially considering its distribution and the morfology of soft tissues. As all contributions from reviewers were considered in this new version of the manuscript, I think it is ready for publication and I reccomend it's acceptance.